# Controlling a robotic arm for functional tasks using a wireless head-joystick: A case study of a child with congenital absence of upper and lower limbs

Sanders Aspelund[1]☺*, Priya Patel[2]☺, Mei-Hua Lee[2], Florian A. Kagerer[2,3], Rajiv Ranganathan[1,2‡], Ranjan Mukherjee[1‡]

1 Department of Mechanical Engineering, Michigan State University, East Lansing, Michigan, United States of America, 2 Department of Kinesiology, Michigan State University, East Lansing, Michigan, United States of America, 3 Neuroscience Program, Michigan State University, East Lansing, Michigan, United States of America

☺ These authors contributed equally to this work.
‡ These authors also contributed equally to this work.
* aspelun1@msu.edu

**Data Availability Statement:** All data are contained in the supporting information: Base Data.xlsx.

## Abstract

Children with movement impairments needing assistive devices for activities of daily living often require novel methods for controlling these devices. Body-machine interfaces, which rely on body movements, are particularly well-suited for children as they are non-invasive and have high signal-to-noise ratios. Here, we examined the use of a head-joystick to enable a child with congenital absence of all four limbs to control a seven degree-of-freedom robotic arm. Head movements were measured with a wireless inertial measurement unit and used to control a robotic arm to perform two functional tasks—a drinking task and a block stacking task. The child practiced these tasks over multiple sessions; a control participant performed the same tasks with a manual joystick. Our results showed that the child was able to successfully perform both tasks, with movement times decreasing by ~40–50% over 6–8 sessions of training. The child's performance with the head-joystick was also comparable to the control participant using a manual joystick. These results demonstrate the potential of using head movements for the control of high degree-of-freedom tasks in children with limited movement repertoire.

## Introduction

According to the 2010 American Census, there were approximately 300,000 children with disabilities requiring some form of assistance with activities of daily living [1]. In this context, assistive devices such as wheelchairs and robotic arms are vital for activities requiring mobility and manipulation. Importantly, these devices are not only critical from a sensorimotor perspective, but they also support psychosocial development by providing children with greater independence [2].

**Funding:** This work was supported by grants from the National Science Foundation (https://www.nsf.gov/) - NSF 1703735 awarded to RM, ML,FK, RR; NSF 1654929 awarded to ML; and NSF 1823889 awarded to RR. The funders had no role in study design, data collection and analysis, decision to publish, or preparation of the manuscript.

Among methods of controlling assistive devices, manual joysticks are the most popular [3]; however, because they require upper limb functionality, they are not suited for individuals with severe motor impairments such as high-level spinal cord injury or congenital limb absence. For these individuals, interfaces have been developed based on signals from the brain [4–6], or the body [7–9]. For children in particular, interfaces based on brain signals, invasive or non-invasive, are less than ideal for long-term use because of issues related to risks of surgery, signal quality, signal drift and longevity [10, 11]. These limitations highlight the need for developing body-machine interfaces that are non-invasive and robust, and, importantly, also have form factors that make them inconspicuous during interaction with peers [12].

A specific class of body-machine interfaces that addresses these requirements are interfaces based on head movements [13]. Head movements are typically preserved in individuals with severe motor impairments, often making such interfaces the only viable option. Two common approaches based on head movements are head arrays and head joysticks. Head arrays rely on a series of switches that are physically activated by contact with the head. Although they are commercially available and have been used for wheelchair control, they are not well-suited for high degree-of-freedom (DOF) tasks because of the binary nature of the switches. Head joysticks, in contrast, mimic manual joysticks and provide a continuous method of controlling degrees of freedom [14, 15], thus having the potential for being used for high-DOF tasks. In addition, head joysticks based on commercially available inertial measurement units (IMUs) are non-invasive, wireless, and have high signal-to-noise ratios. Previous research has shown the utility of head joysticks for low-DOF tasks such as wheelchair control [14, 16, 17], but evidence of control of high-DOF tasks using head joysticks is limited [18, 19], especially in children.

In this study, we investigated the use of an IMU-based head joystick for controlling a robotic arm to perform high-DOF functional tasks. In a child with congenital absence of all four limbs, we examined the child's ability to perform two tasks related to activities of daily living–(i) picking up a cup and drinking using a straw, and (ii) manipulating objects placed on a table. We show that the child can use the head-joystick to successfully perform these complex tasks and improve over time to a level that is comparable to that of an unimpaired individual using a manual joystick.

## Materials and methods

### Participants

Our main participant was a 14-year old male with congenital absence of all four limbs—see Fig 1a. He had participated in two previous studies with our group which involved position control of a cursor [20] and 2-DOF velocity control of the end-effector of a robotic arm [21]. These prior studies involved the control of these devices using shoulder and torso movements. In the current study, he used his head as a joystick to control the robotic arm as shown in Fig 1a. Initially, there were 4 unstructured sessions, each lasting about 30–45 minutes in length. We used these sessions to calibrate the interface to make sure that the head movements performed were in a comfortable range when controlling the robot. During each session, the participant was asked to do some exploratory movements of the head to understand how movements of each DOF controlled the robot, and to learn the operation of the switches (which were used to toggle between the translation/orientation modes and control the end-effector). In addition, the participant was also free to perform any tasks of their liking using the robot arm like trying to pick up an object from a table. The child was paid $10 per visit.

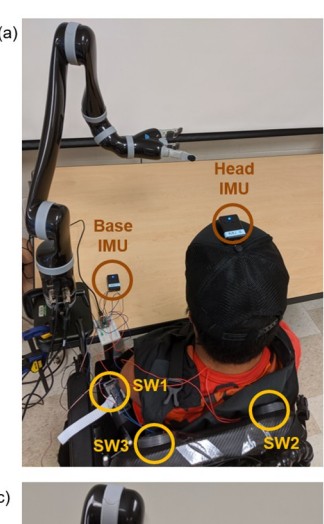
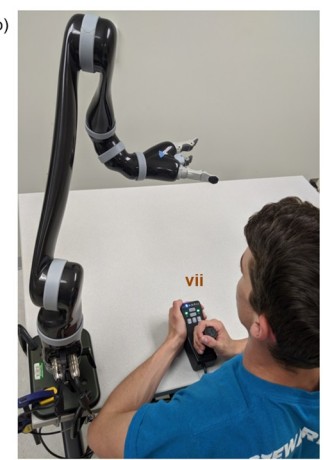
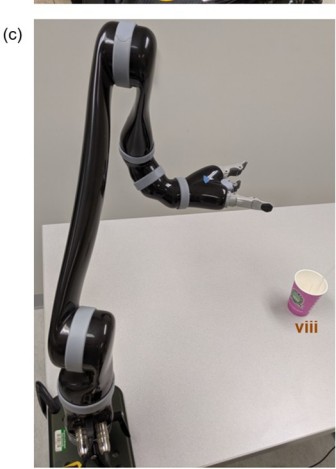
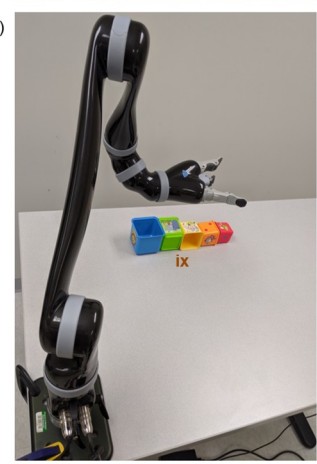

**Fig 1. Interfaces for controlling the robot and experimental setup for the drinking and stacking tasks.** (a) Interface for main participant using the head-joystick. A head mounted IMU was used to control the robotic arm, and switches (SW1, SW2, SW3) placed behind the shoulder were used to toggle between different control modes and for control of the grasper. (b) Interface for control participant using the manual joystick. (c) Initial layout of drinking task. Participants had to use the robotic arm to grasp the cup and bring it to the mouth. (d) Initial layout of stacking task. Participants had to use the robotic arm to stack the five blocks on top of each other in order of decreasing size with the biggest block at the base.

Our control participant was an able-bodied college-aged male volunteer (21 years old)- see Fig 1b. He controlled the robot with its accompanying manual joystick. He had no prior experience interacting with the system or observing its use.

All participants provided informed consent or assent (including parental consent in case of child) and experimental protocols were approved by the IRB at Michigan State University. The individuals pictured in Fig 1 (and the S2 and S1 Videos) have provided written informed consent (or parental consent when appropriate, as outlined in PLOS consent form) to publish their images and videos alongside the manuscript.

## Apparatus

Robot: We used a 7-DOF robotic arm (JACO v2 arm, KINOVA robotics, Boisbriand QC, Canada) mounted on a table for performing the object manipulation tasks. The robotic arm, shown in Fig 1, is anthropomorphic with 2 DOFs at the shoulder, 1 DOF at the elbow,

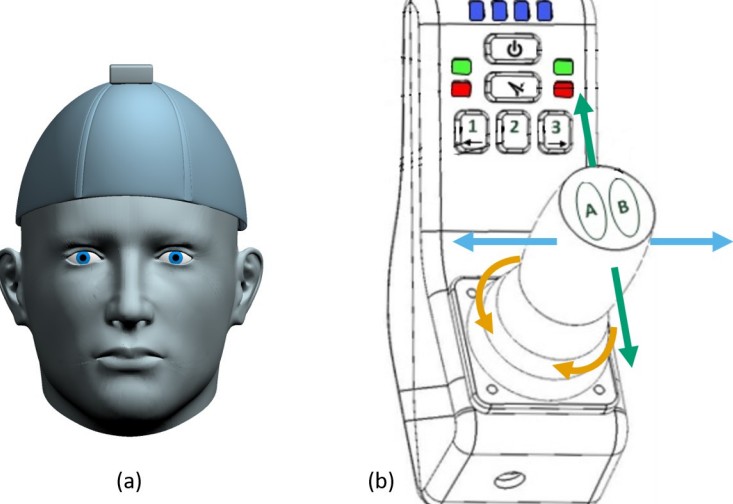

**Fig 2. Configurations of head-joystick and manual joystick.** (a) Configuration of the head-joystick. It consisted of a three DOF wireless inertial measurement unit (IMU) (YEI Technologies Inc.) on top of a baseball cap with the bill removed. (b) Configuration of the manual joystick. The joystick could move forward or backward, left or right, or be twisted clockwise or counterclockwise. Buttons on the joystick enabled the user to switch control modes as indicated by the light-emitting diodes (LEDs) at the top.

3 DOFs at the wrist, and a 1 DOF gripper; specifications of the robot can be found at www.kinovarobotics.com.

Head-Joystick: For the child with congenital limb absence, the robotic arm was controlled via signals generated by a wireless inertial measurement unit (IMU) (YEI Technologies Inc., Ohio) worn on the top of a baseball cap with the bill removed—see Figs 1a and 2a. A second IMU, shown in Fig 1a, was placed on the table to determine the relative orientation between the participant and the robot reference frame. This second IMU, although redundant in the current study (because the table was always fixed), is necessary for preventing unintended movement of the robotic arm when the robot reference frame moves along with the participant (for example, when the robot is mounted on the wheelchair of the participant). Together, these IMUs were used to control the six DOFs of the robot end-effector—three position DOFs and three orientation DOFs. The head-joystick was not used to control the seventh DOF—opening and closing of the end-effector, which was performed by operation of switches described below.

Switches: In addition to the two IMUs, there were three switches. The first switch (SW1) enabled toggling between position control of the end-effector, orientation control of the end-effector (both using the head-joystick), and a no-movement mode. SW1 was a small button-type off-the-shelf switch (requiring a 1 N activation force) placed below the primary participant's left shoulder—see Fig 1a). The three modes allowed the participant to use the head-joystick to control all six DOFs of the robot, and the no-movement mode allowed the participant to freely move his body when not intending to control the robot—see Fig 3a. Two additional switches (SW2 and SW3—see Fig 1a) were attached to the chair's backrest, behind the shoulders of the participant, and controlled the opening and closing of the end-effector. These switches were custom-made with a diameter of 70 mm, a throw of 1 mm, and an activation force of approximately 10 N. Pressing only SW2 caused the grasper to close, while pressing only SW3 caused it to open. Pressing neither or both switches resulted in the current grasp

Fig 3. Description of switches used to toggle between modes. (a) The three modes of end-effector control as toggled by switch one (SW1). (b) Switches SW2 and SW3 were used to close and open the grasper. The truth table shows how pressing and releasing each button on and off causes the grasper to react.

being maintained—see Fig 3b. The participant was able to determine the state of the grasper from LEDs placed on the table—see Fig 1a.

Traditional Joystick: The able-bodied adult participant controlled the robotic arm using the manual joystick (shown in Figs 1b and 2b). This allowed for 3-DOF end-effector position control, 3-DOF end-effector orientation control, and 1-DOF opening and closing of the end-effector grasper. These three modes were toggled using buttons on the joystick while LEDs (located on the joystick) signaled the active mode. This functionality inspired the design of the head-joystick and switches used by our primary participant.

## Controlling the six DOFs of the robot using the head-joystick

The head-joystick has three independent DOFs associated with the head tilting up and down (neck flexion/extension, see Fig 4a), the head turning right and left (rotation, see Fig 4b), and the head tilting right and left (lateral flexion/extension, see Fig 4c). These three DOFs were measured by the IMU on the head and mapped to control either the three DOFs of the end-effector position or the three DOFs of the end-effector orientation. We used a velocity-control mode where the IMU signals were mapped to velocity commands. The robot internally performed inverse kinematic and inverse dynamic computations for joint angle velocities and actuator torques to produce the commanded end-effector velocities. Velocity commands for the end-effector position were computed with respect to the base frame of the robot (XYZ frame in Fig 4d), while velocity commands for the end-effector orientation were computed with respect to the body-fixed frame of the end-effector ($e_1e_2e_3$ frame in Fig 4e).

In the end-effector position control mode (achieved by toggling SW1), tilting the head upwards relative to a neutral home head orientation, as shown in Fig 4a, resulted in an upward (+Z) motion of the end-effector—see Fig 4d. Through the same process, tilting the head downwards (see Fig 4a) resulted in downward motion (-Z). The magnitude of the velocity command was proportional to the angle of head tilt. Similarly, turning the head right and left (see Fig 4b) resulted in the end-effector motion towards the right (+X) and left (-X), respectively—see Fig 4d. Finally, following the right-hand-rule, tilting the head to the right and left (see Fig 4c) resulted in end-effector movement forward (+Y) and backwards (-Y)—see Fig 4d.

In the end-effector orientation control mode (achieved by toggling SW1—see Fig 3a), the IMU signals were translated to rotational velocities of the end-effector about its body-fixed frame. Tilting the head up and down (see Fig 4a) resulted in the end-effector pitching about its $e_1$ axis in the positive and negative direction (see Fig 4e). Similarly, turning the head right and left (see Fig 4b) resulted in the end-effector rotating about its $e_3$ axis in the negative and

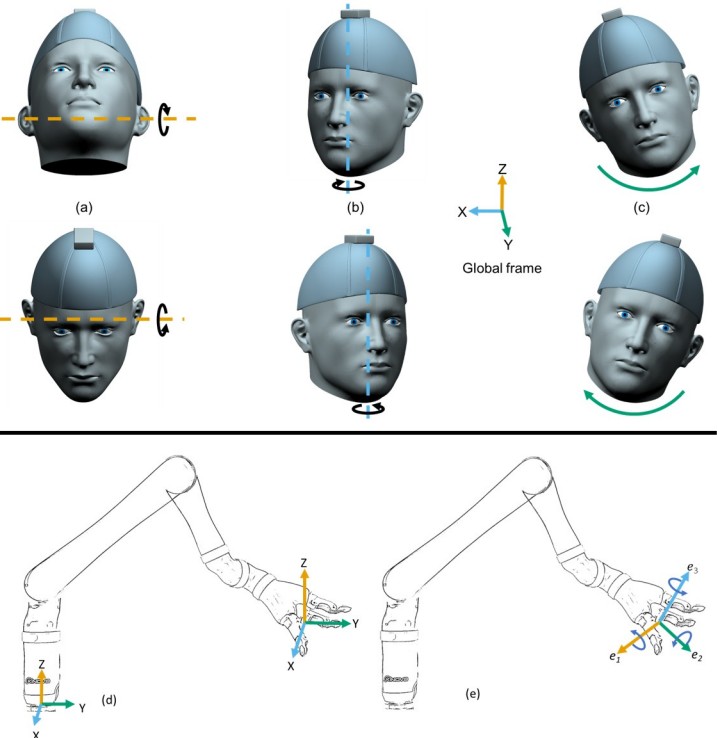

**Fig 4. Mapping between head motion and robotic arm motion for the head joystick.** Three sets of possible motions of the head for controlling the three DOFs of the head-joystick: (a) tilting the head up and down, (b) turning the head right and left, (c) tilting the head right and left. (d) Coordinate frame fixed to the robot base for controlling the position of the end-effector showing the corresponding change in position of the end-effector for each set of head motions. (e) Coordinate frame fixed to the robot end-effector for controlling the orientation of the end-effector showing the corresponding orientation of the end-effector for each set of head motions.

positive direction. Finally, tilting the head to the right and left (see Fig 4c) caused the end-effector to rotate about the $e_2$ axis in the positive and negative direction, respectively.

Because even small unintentional deviations from the resting posture could be captured by the IMUs and potentially affect the velocities, we implemented a 'dead-zone' of 0.1 radians ($\approx$6 deg) so that robot actually started moving only when the IMU roll, pitch, or yaw angles exceeded this threshold. When a measured angle exceeded this threshold, we subtracted the value of 0.1 from the magnitude of angle before computing the commanded velocity to maintain smooth control (e.g. an IMU yaw angle of 0.15 rad would only cause the end-effector to move at 0.05 m/s to the X direction in cartesian mode). The dead-zone not only provided the user with a larger range of rest postures, but also helped the user generate distinct commands along a single direction as velocities in the other two directions would be under the threshold.

## Tasks

We used two tasks that mimicked activities of daily living to assess the participants' abilities to control all 7 DOFs of the robotic arm: a drinking task and a stacking task.

**Drinking task.** The first task involved drinking from a cup. The participant was required, as quickly as possible, to reach for and grasp a cup containing liquid, bring the cup towards the mouth, and drink from it using a straw. The paper cup was semi-rigid: it provided enough resistance to be firmly grasped while also deforming enough to allow the participant to visually confirm the strength of the grasp. The robotic arm was always initialized in the same starting

position (X: 0.40 m, Y: 0.30 m, Z: 0.30 m from the base of the robot with the gripper open enough to grasp the cup and facing towards the right) while the cup was in front of and to the right of the participant on the table (mean X: 0.58 m, mean Y: 0.25 m, Z: 0.00 m)—see Fig 1c. The straw was also kept consistently in the same orientation. To impose similar constraints on both participants, the control participant was instructed to minimize trunk movement and to move the straw to their mouth (and not move their upper body towards the straw). The final position of the grasper holding the cup when the main participant took a drink was (mean X: 0.35 m, mean Y: -0.04 m, Z: 0.10 m) with the grasper facing towards the participant.

**Stacking task.** The second task involved stacking five cube-shaped blocks of decreasing size (see Table 1), each with an open face, on top of each other—see Fig 1d. From the same initial end-effector position as the drinking task, each block had to be grasped, reoriented, and placed on the previous larger block to build a tower. Each block had a lip that protruded 8–10 mm outside the base of the next larger block, thereby determining the required accuracy needed to successfully stack the blocks. The location of the tower was chosen by the participant (mean X: 0.39 m, mean Y: 0.28 m) to allow them to have a clear vision of the remaining blocks for the remainder of the tasks.

The blocks were placed directly across the table from the participant in five orientations not matching the target orientation—see Fig 1d. The first block had its opening facing upwards; the second block had its opening facing away from the participant; the third block had its opening facing towards from the participant; the fourth block had its opening facing to the right of the participant; the fifth block had its opening facing to the left of the participant. These starting positions were standardized throughout trials (position of first block with respect to the base of the robot: mean X: 0.31 m, mean Y: 0.49 m). Each block required a different approach strategy and subsequent placement strategy. Indeed, this task was more difficult than the drinking task.

## Protocol

**Child with congenital limb absence.** The amount and distribution of practice for both participants are shown in Table 2. We made nine visits to the school of the main participant for testing him over a period of 2 months. Visits were during the participant's free period and were not evenly spaced as they were subject to scheduling availability such as school breaks and exams. Each session was no longer than 60 minutes as constrained by the participant's class schedule. He performed a total of 19 drinking task trials and 11 stacking task trials over the course of all the sessions. The number of trials during each session was variable and

**Table 1. Dimensions of five blocks used in the stacking task.**

| Block | Outer dimension of closed face (mm) | Inner dimension of open face (mm) |
|---|---|---|
| 1 | 67 | 84 |
| 2 | 60 | 77 |
| 3 | 53 | 68 |
| 4 | 46 | 61 |
| 5 | 44 | 54 |

The difference between the outer dimension of the previous block and the inner dimension of the subsequent block defined the precision to which the block had to be placed to be secure. For example, the precision requirement when stacking the 2nd block on top of the 1st was 77–67 = 10 mm, and that for stacking the 5th on the 4th block is 54–46 = 8 mm.

**Table 2. Experimental protocol.**

| | MAIN PARTICIPANT | | | | | | | | | CONTROL PARTICIPANT | | | |
|---|---|---|---|---|---|---|---|---|---|---|---|---|---|
| **Visit** | 1 | 2 | 3 | 4 | 5 | 6 | 7 | 8 | 9 | 1 | 2 | 3 | 4 |
| **Day** | 1 | 4 | 8 | 11 | 45 | 50 | 52 | 57 | 64 | 1 | 5 | 8 | 12 |
| **Drinking task trials** | 2 | 7 | 0 | 0 | 5 | 4 | 0 | 1 | 0 | 2 | 7 | 0 | 2 |
| **Stacking task trials** | 0 | 0 | 1 | 3 | 1 | 0 | 2 | 2 | 2 | 0 | 2 | 3 | 3 |

Experimental protocol showing the amount and distribution of practice across days for the main participant (i.e. the child with congenital limb absence) and the control participant. For the main participant, there were a total of 19 trials on the drinking task and 11 trials on the stacking task. For the control participant, there were a total of 11 trials on the drinking task and 8 trials on the stacking task.

dependent on the tasks performed: a stacking task generally took longer than the drinking task. For both participants, the study was concluded when their performance plateaued on each task.

During each session, the main participant sat in his personal wheelchair with the table at his navel level, robot to his left front side and eyes at the level of robot's shoulder joint. He wore a cap with an IMU attached on top of it. The IMUs sampled at 125Hz, the same rate at which the signals were processed and sent as commands to the JACO arm. The states of both the head IMU and chair IMU were polled continuously and their difference was sent to the robotic arm as either cartesian velocity commands or rotational velocity commands, depending on the current mode. All IMU values were taken relative to a comfortable base position as defined by the participant before starting each trial.

**Control participant.** There were 4 lab visits made by the control participant over a two-week period (2 visits per week). Each session was no longer than 60 minutes to match the main participant's sessions. A total of 10 drinking task trials and 8 stacking task trials were performed over the course of all the sessions. Similar to the main participant's sessions, the number of trials per session varied and were dependent on the tasks performed.

The control participant sat at a table with the height adjusted such that the participant's mouth was at the same height relative to the table's surface as the main participant's. This was to ensure the movement domain of the robotic arm would be similar between participants. This participant was instructed to not translate his head location significantly during the tasks as this could provide an unfair advantage relative to the main participant.

It is important to note that the control participant was an adult controlling a manual joystick; therefore, these data are not intended to be a direct comparison with the child using the head-joystick. Rather, given that the tasks we used were complex tasks for which benchmarks are not already available, the data from the control participant provide reference values that help in interpretation of the magnitudes of the change in performance with learning and the final performance level achieved by the child.

## Procedure

At the beginning of each session, participants were allowed to explore the robotic arm's range of movement for as long as they wanted. This generally lasted anywhere between 3–5 minutes. The goal of this free exploration was just to ensure that the interface was working as intended and the participant was ready to start controlling the robot arm.

The order of tasks was decided based on discussion with the participant. The robotic arm was put in a home position before every trial. Breaks were taken between trials if the participant wanted to.

The experimenters also occasionally provided 'coaching' in the form of suggested movements and grasping strategies during both trials and breaks. The type and amount of coaching was not predetermined as the goal of this study was to determine the best level of performance capable with the interface. The amount of coaching decreased over time as control skill and strategies improved.

## Data analysis

### Task completion

Task completion was measured by the number of successful trials at the task. For the stacking task, trials were considered incomplete if the participant was not able to finish stacking all five blocks. However, we still report the characteristics of these incomplete trials in the data analysis as they potentially reflect exploration and learning strategies.

### Movement time

Movement time for the tasks were computed from video recordings of the session. For the drinking task, the movement time began on the frame when the robot was first moved from its initial home position by the participant and ended when the participant's mouth made contact with the straw. For the stacking task, the movement times were split into movement times for each block. The first block's time began on the frame when the robot first moved and ended when the robot was no longer touching the correctly placed block. The next block's time began when the previous block's time ended. Together these were combined into a total completion time for the block stacking task.

### Dimensionless jerk

To quantify the smoothness of the movement, we computed the dimensionless jerk values for the tasks from the end-effector position data on each trial. The jerk was normalized by the time and peak velocity to yield a dimensionless measure which has been shown to be more appropriate measure [22]. The position values were first low pass filtered using a $2^{nd}$ order Butterworth filter with a cutoff frequency of 6 Hz. The jerk values for each trial were then calculated from subsequent derivatives of the filtered position data and integrated over the duration of the trial. The dimensionless jerk value was then computed as follows

$$Dimensionless\ Jerk = \sqrt{\left(\int_{t_1}^{t_2} \|\dddot{x}\|^2 dt\right) MT^3 / v_{peak}^2}$$

where $\dddot{x}$ indicates the instantaneous jerk, $MT = (t_2 - t_1)$ is the movement time of the trial, and $v_{peak}$ indicates the magnitude of the peak velocity during the trial.

## Results

### Task completion

For the drinking task, both participants completed all trials (main participant = 19/19 trials; control participant = 11/11 trials).

For the stacking task, the main participant completed 8/11 trials. Three stacking trials were considered incomplete because the final stacking block (red block in Fig 1d) was dropped to a

position outside of the reach of the robotic arm during the course of the trial. The control participant completed 8/8 stacking trials. Examples of the main participant performing the two tasks are shown in S1 and S2 Videos.

## Movement time

**Drinking task.** The main participant's slowest movement time was measured at 99 s, his fastest time was measured at 30 s, while the average of his movement times was 55.2 s (SD = 16.3 s) (see Fig 5). However, two trials on visit #5 (trials marked with an "x" on Fig 5) involved a lot of talking during the trial and a poorly aligned IMU. Excluding these two trials showed the main participant improved by 43% from his first trial. The control participant's slowest movement time was 75.1 s, his fastest time was 16.9 s, while the average of his movement times was 36.0 s (SD = 15.7 s). Using the head joystick, the main participant's average movement time was around 40% slower than the control participant's average movement time when using the manual joystick.

**Stacking task.** For completed trials, the main participant's slowest movement time was 1020 s, his fastest time was 446 s, while the average of his movement times was 608 s (SD = 175 s) (see Fig 6). The control participant's slowest movement time was 797 s, his fastest time was

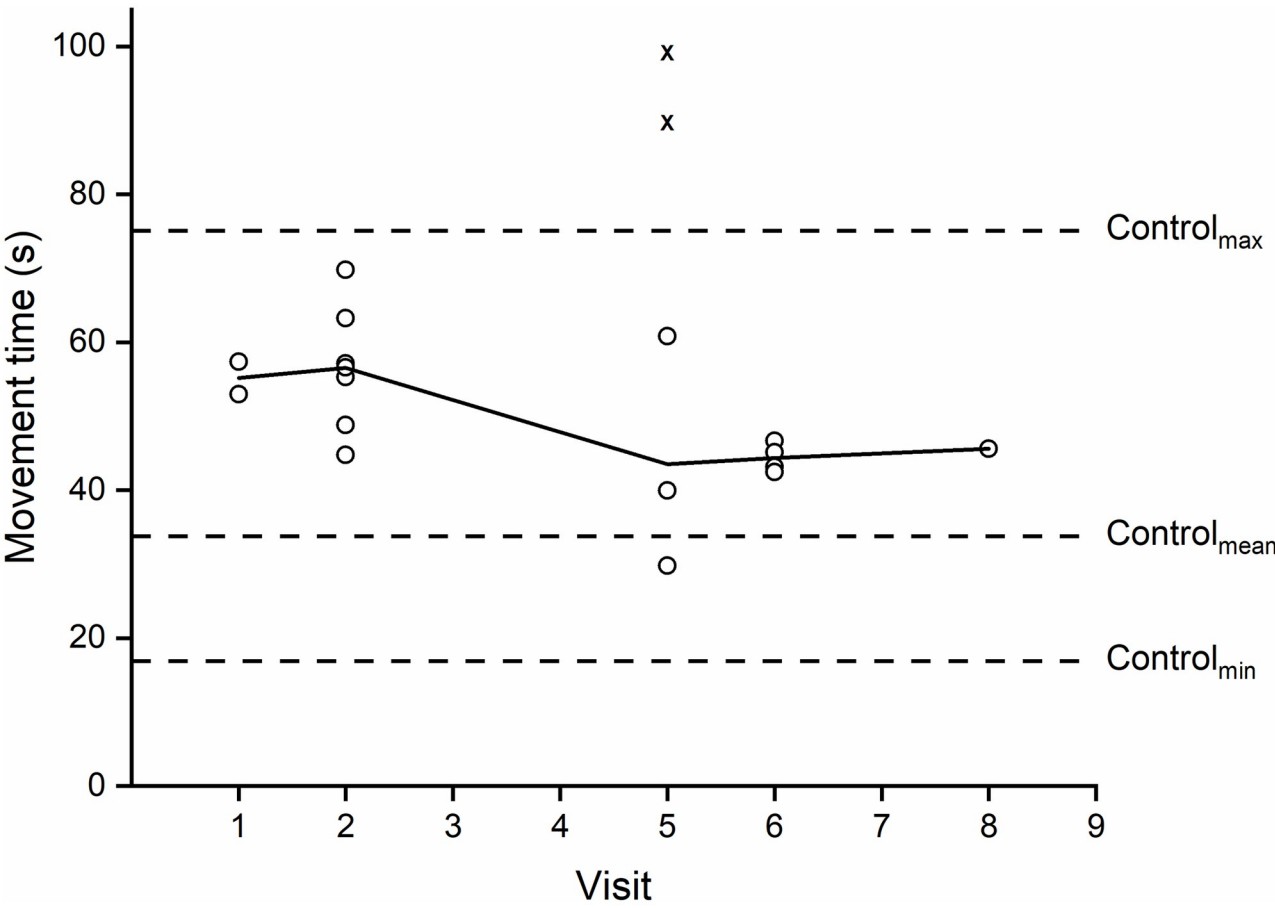

**Fig 5. Movement times for drinking task trials.** The movement times for the drinking task trials for the main participant across practice. The range and mean of the control participant's times are included for comparison. Movement times denoted with a "x" were those in which the main participant had significant distractions or poor IMU alignment. They are not included in mean session movement time but are reported since the main participant was able to complete the task.

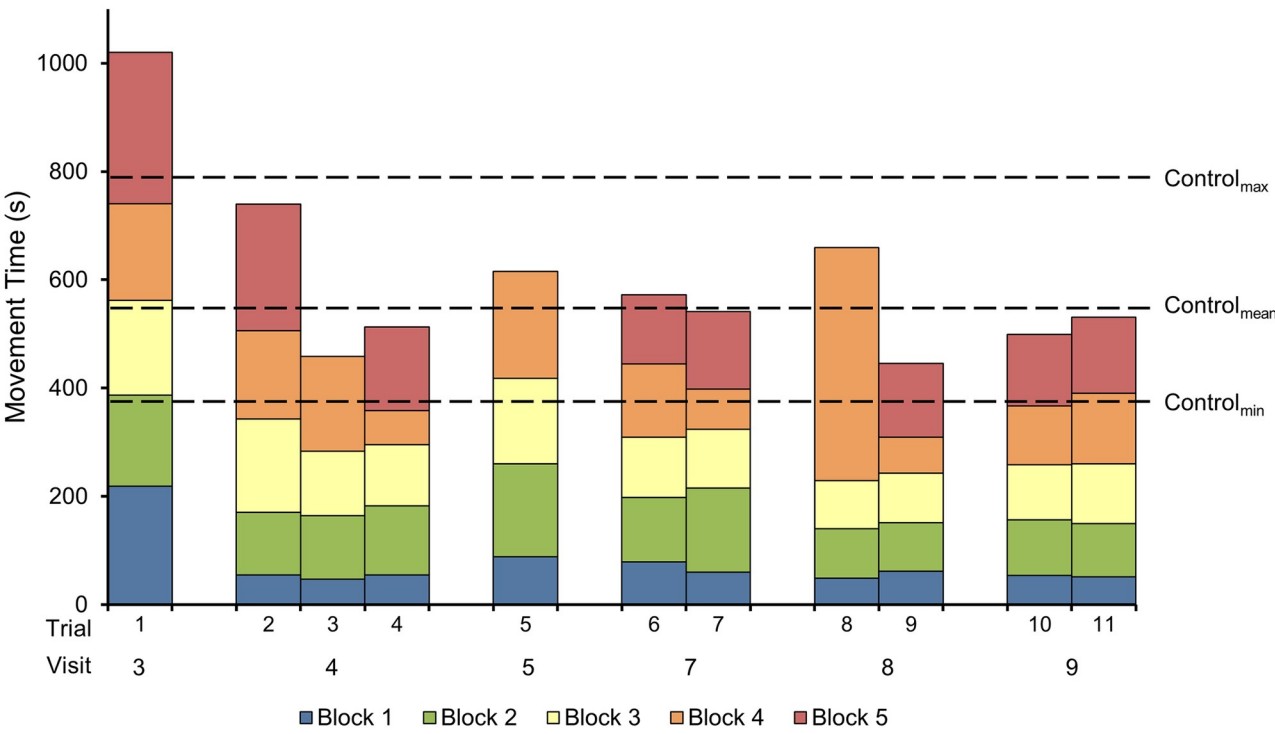

**Fig 6. Movement times for stacking task trials.** The movement times for the stacking task for the main participant across practice. The range and mean of the control participant's complete stacking times are included for comparison. Trials 3, 5, and 8 were incomplete in which the main participant was unable to successfully place the last block due to dropping it outside the reach of robotic arm.

376 s, while the average of his movement times was 549 s (*SD* = 128 s). Overall the reduction in their completion times on their last days' trials relative to their respective first trials were very comparable (main participant = 50%; control participant = 39%) and the main participant's best trial was only 19% slower than the control participant's best trial. For the main participant, the average times it took to successfully place each block were 74 s (*SD* = 47 s), 123 s (*SD* = 28 s), 123 s (*SD* = 30 s), 156 s (*SD* = 98 s), and 168 s (*SD* = 52 s), in order. Average improvements in movement time on the last day of each block relative to trial 1 were 76%, 40%, 39%, 33%, and 51%.

## Dimensionless jerk

**Drinking task.** The dimensionless jerk of the main participant closely followed the pattern of the movement times—see Fig 7. Two trials on visit #5 (trials marked with an "x" on Fig 5) involved a lot of talking during the trial and a poorly aligned IMU. Additionally, the data for trial #15, which occurred during visit 6, was corrupted and therefore omitted from the jerk analysis. Comparing the dimensionless jerk values from the first four trials (Trials 1–4) to the last four trials (Trials 16–19)) shows a 45% decrease in dimensionless jerk value.

**Stacking task.** The dimensionless jerk of the main participant for the stacking trials also closely followed the pattern of the movement times—see Fig 8. Comparing the jerk values from the first three completed trials (trial 3 (visit 4), trial 5 (visit 5), and trial 8 (visit 8) were incomplete) to the last three completed trials shows an 57% decrease in dimensionless jerk value.

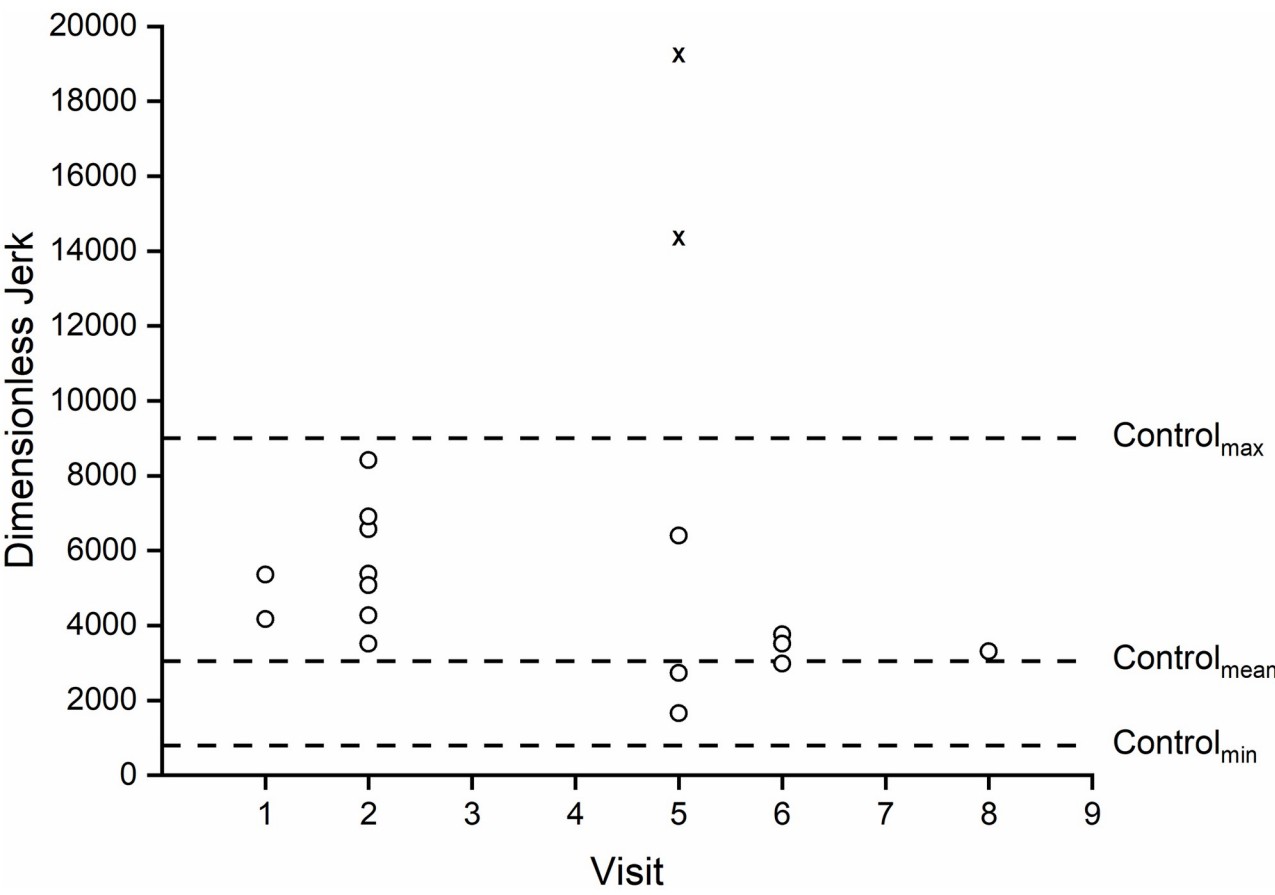

**Fig 7. Dimensionless jerk values for drinking task trials.** Dimensionless jerk values for drinking task trials for the main participant across practice (smaller jerk values indicate smoother movements). Jerk values denoted with a "x" were those in which the main participant had significant distractions or poor IMU alignment which led to large jerk values. Additionally, trial #15, which occurred during visit 6, is omitted due to data corruption.

## Discussion

The goal of this study was to examine the use of an IMU-based head-joystick for controlling a robotic arm to perform high-DOF functional tasks. We showed that a child with congenital limb absence was able to successfully use the head-joystick to perform two complex functional tasks. Moreover, the child was able to improve his performance over time to a level comparable to that of an unimpaired individual using a manual joystick.

Across a fairly limited practice time (~6–8 sessions) for both the drinking and stacking tasks, the child achieved the task goal almost twice as fast as compared to his first attempt. These times were, as expected, somewhat higher relative to the performance of the control participant on the joystick, but here we found an effect of task complexity: in the simpler drinking task, the performance of the child was about 40% slower than the control, whereas in the more complex stacking task, this difference shrunk to about 20%. A likely explanation for this is that even though the control algorithms were identical in both cases, in the simpler drinking task, where the robot could travel with higher velocity, the manual joystick had an advantage because the user could simply push the joystick instantaneously to the end of its range of motion and maintain it in that spot without discomfort. In contrast, such rapid movements would have been difficult using the head. One alternative could have been to increase the gain

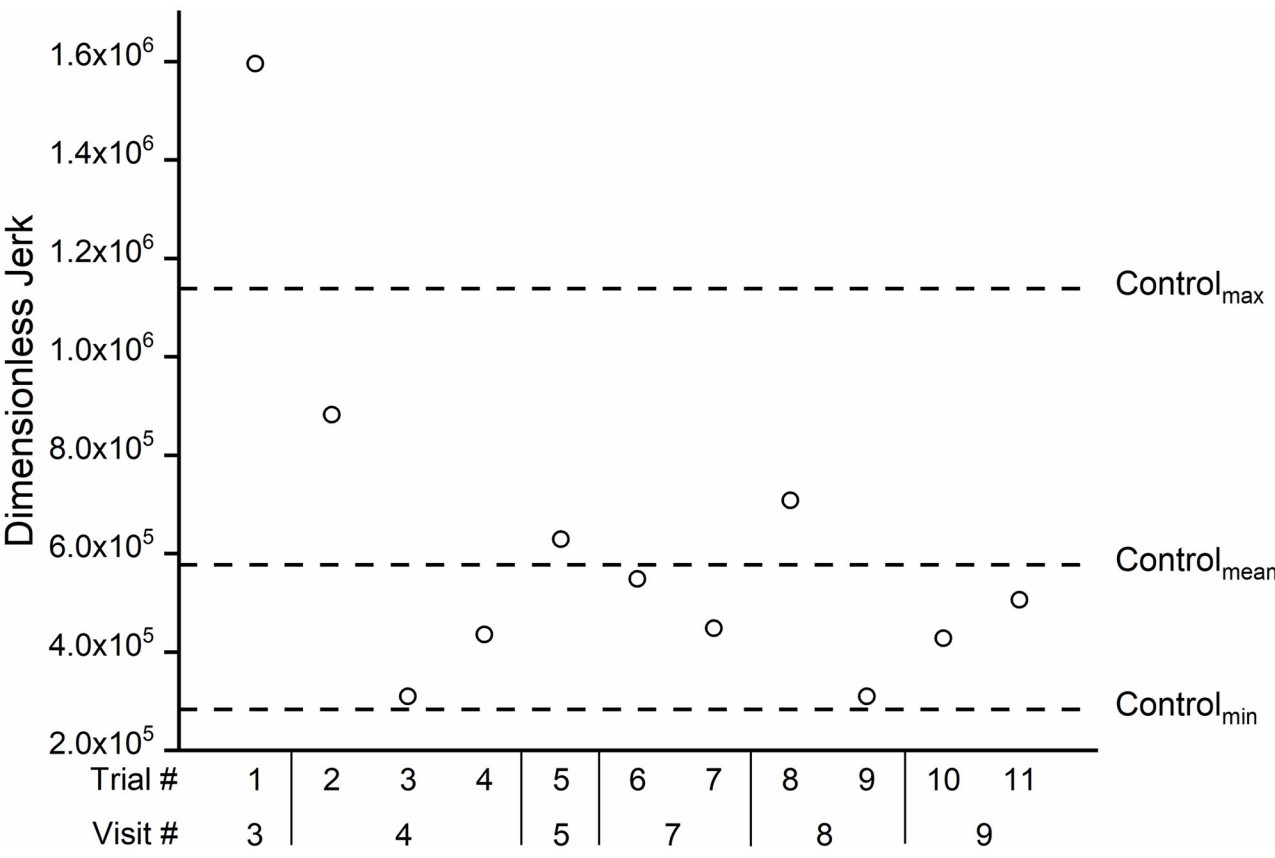

**Fig 8. Dimensionless jerk values for stacking task trials.** Dimensionless jerk values for stacking task trials for the main participant across practice (smaller jerk values indicate smoother movements). Trial 3 (visit 4), trials 5 (visit 5) and 8 (visit 8) were incomplete in which the main participant was unable to successfully place the last block due to dropping it outside the reach of robotic arm.

on the head joystick, but this would have likely compromised fine control required in more complex tasks. However, in the more complex stacking task, where movement speed was not the limiting factor in performance, the head-joystick was almost on par with the manual joystick. Moreover, although we had no direct measures of user satisfaction, the fact that the participant continued this task for over 3 months and was enthusiastic about returning for future visits is a potential indicator that he was satisfied with the interface.

Controlling high DOFs using a body-machine interface in individuals with limited movement repertoire has always posed a significant challenge. One popular approach is to use dimensionality reduction techniques like principal component analysis (PCA) to extract the most relevant movement directions for control. While this technique can accommodate different movement repertoires, they have only been implemented for controlling one or two degrees of freedom [7, 9]; furthermore, the mapping between the motion of the body and that of the assistive device can often be non-intuitive [21]. A more recent approach—the Virtual Body Model (VBM) [23]- is more intuitive for control of high DOFs because of the pre-defined mapping between the body and device DOFs but it relies on a nearly full range of movement in the torso. Since our primary participant had no lower limbs, seatbelts are used to constrain his body to an upright posture in the wheelchair; this limited range of movement of the torso makes the VBM approach unsuitable in our case. These specific constraints required the design of a custom interface that relied primarily on head movements. In this design, the head

was used as a joystick to control up to three DOFs of the end-effector at a time; toggling between different sets of DOFs was achieved by activating switches using the body.

One of our primary goals was to provide greater independence for the child in activities of daily living. To this end, we designed the two tasks to not only involve control of high DOFs, but also resemble activities that are frequently needed in both the home and school environments. The drinking task required the child to position the robot end-effector near the cup, grasp the cup, and position and orient the cup near his mouth so that he could drink comfortably using the straw. The stacking task was more complex; it not only involved proper positioning and orienting of the end-effector to grasp and place individual blocks, but also required sequence planning and on-the-fly adaptations to accommodate for variations in prior movement outcomes. For example, the final block of the task required a two-step strategy: given the distant location and orientation of the block, it could not be grasped and placed on the stack using a single grasp. Instead, the block needed to be repositioned and released before regrasping it in such a way that it could be placed on the stack. The use of such tasks with several levels of complexity may especially be critical when designing interfaces of children, as performance may not only be determined by the intuitiveness of the control but also the cognitive planning of the task. It is also worth noting that despite requiring the use of head movements to control the robot to precisely control the end effector, the child was able to successfully perform these tasks, indicating that the small head movements performed did not interfere greatly with the use of visual feedback.

Our work also extends prior work on using head gestures to control a robot. In one study [18], adult participants (able-bodied adults and tetraplegics) used a similar head-mounted IMU to control a 7 DOF robotic arm to accomplish pick and place tasks, but focused only on a single session of practice. Similarly, a second study [19] evaluated the use of IMUs for the control of a robot arm, and showed similar performance in a single session in able-bodied adults. Our results add to these prior findings by demonstrating that (i) these interfaces are well-suited for children, and (ii) the improvement in performance over multiple practice sessions is substantial (up to 40–50% reduction in movement times). The child's performance for the tasks was found to be comparable to that of an adult control participant using a manual joystick. However, given that we only had data from a single child and a single adult, additional studies are needed for assessing the generality of these findings.

In addition to head control methods discussed here, several alternate control interfaces to manual joysticks have been developed for individuals with severe movement impairments. These include sip-and-puff systems, voice control [24], gaze control [25] and tongue control [26]. These interfaces typically involve some tradeoff between (i) the number of control dimensions (e.g., a device that only allows control of 1 or 2 dimensions would require frequent 'switching' to control a high DOF robotic arm), (ii) the type of control (e.g., discrete controls are possible using voice commands but are less intuitive and precise relative to continuous control like a joystick), and (iii) the 'invasiveness' of the device both in terms of its physical attributes (e.g., whether it is easily wearable, wireless etc.) but also how it affects other activities such as communication (e.g., gaze or voice based controls may interfere with natural day-to-day behavior). Ultimately, the choice of the interface will depend both on existing movement abilities for the individual and the number of degrees of freedom to be controlled.

In terms of further improvements to our design, we wish to highlight two issues. First, a limitation of our approach is that the 'burden of learning' is all on the user. This may be especially challenging for children, who show deficits relative to adults in learning such interfaces [21, 27]. One way to improve this is to use either an adaptive interface that adjusts to the user [28, 29], or use a shared control framework so that the autonomy of control can be shared between the human and the machine [30]. Second, for the sake of simplicity, we relied only on

head movements (i.e. kinematics) to control the device. However, in the control of other neuroprosthetics, electromyographic signals from different muscles is often used to augment the movement repertoire by providing distinct control signals for the control of the external device [31–33]. Therefore, a hybrid combination of IMU signals along with electromyography may further facilitate the user for efficient control of high DOFs [34]. Addressing these limitations could increase the potential of this approach to deal with real-life situations which require both speed and accuracy.

In conclusion, we showed that for a child with congenital limb absence, a head-joystick is a viable means for controlling a robotic arm to perform complex tasks of daily living. Developing efficient, non-invasive techniques with intuitive control of high DOFs, and quantifying their performance in a larger sample is a key challenge that needs to be addressed in future studies.

## Supporting information

**S1 Video. Example drinking task for main participant.** In this trial, the main participant conducts a complete drinking task. The robotic arm is initialized to the starting position after which the participant commands it to move towards the cup, grasp the cup, and then move and orient the cup such that he is able to drink from the straw.
(MP4)

**S2 Video. Example stacking task for main participant.** In this trial, the main participant conducts a complete stacking task. The robotic arm is initialized to the starting position after which the participant commands it to approach, grasp, orient, and place each block on the preceding block or on the table in the case of the first block.
(MP4)

**S1 Data. Data from participant trials as used for analysis.** Included are the movement times and dimensionless jerk values for each of the main and control participants' drinking and stacking trials.
(XLSX)

## Author Contributions

**Conceptualization:** Sanders Aspelund, Mei-Hua Lee, Florian A. Kagerer, Rajiv Ranganathan, Ranjan Mukherjee.

**Formal analysis:** Sanders Aspelund, Priya Patel.

**Funding acquisition:** Mei-Hua Lee, Florian A. Kagerer, Rajiv Ranganathan, Ranjan Mukherjee.

**Investigation:** Sanders Aspelund, Priya Patel.

**Methodology:** Sanders Aspelund, Priya Patel, Mei-Hua Lee, Florian A. Kagerer, Rajiv Ranganathan, Ranjan Mukherjee.

**Software:** Sanders Aspelund.

**Supervision:** Rajiv Ranganathan, Ranjan Mukherjee.

**Visualization:** Sanders Aspelund, Priya Patel.

**Writing – original draft:** Sanders Aspelund, Priya Patel, Florian A. Kagerer, Rajiv Ranganathan, Ranjan Mukherjee.

**Writing – review & editing:** Sanders Aspelund, Priya Patel, Mei-Hua Lee, Florian A. Kagerer, Rajiv Ranganathan, Ranjan Mukherjee.

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
