## [Decision Letter · Decision Letter 0]

30 Jan 2020

PONE-D-19-31845

Controlling a robotic arm for functional tasks using a wireless head-joystick: A case
study of a child with congenital absence of upper and lower limbs

PLOS ONE

Dear Mr. Aspelund,

Thank you for submitting your manuscript to PLOS ONE. After careful consideration, we
feel that it has merit but does not fully meet PLOS ONE’s publication criteria as it
currently stands. Therefore, we invite you to submit a revised version of the
manuscript that addresses the points raised during the review process.

We would appreciate receiving your revised manuscript by Mar 14 2020 11:59PM. When
you are ready to submit your revision, log on to https://www.editorialmanager.com/pone/ and select the 'Submissions
Needing Revision' folder to locate your manuscript file.

If you would like to make changes to your financial disclosure, please include your
updated statement in your cover letter.

To enhance the reproducibility of your results, we recommend that if applicable you
deposit your laboratory protocols in protocols.io, where a protocol can be assigned
its own identifier (DOI) such that it can be cited independently in the future. For
instructions see: http://journals.plos.org/plosone/s/submission-guidelines#loc-laboratory-protocols

We look forward to receiving your revised manuscript.

Kind regards,

Imre Cikajlo, Ph.D.

Academic Editor

PLOS ONE

Journal Requirements:

3. We note that Figure 1 and your videos includes an image of a
[patient / participant / in the
study]. 

4. We note you have included a table to which you do not refer in the text of your
manuscript. Please ensure that you refer to Table 1 in your text; if accepted,
production will need this reference to link the reader to the Table.

Reviewers' comments:

Reviewer's Responses to Questions

**Comments to the Author**

1. Is the manuscript technically sound, and do the data support the conclusions?

Reviewer #1: Yes

Reviewer #2: Partly

Reviewer #3: Partly

2. Has the statistical analysis been performed
appropriately and rigorously? 

Reviewer #1: N/A

Reviewer #2: N/A

Reviewer #3: N/A

3. Have the authors made all data underlying the
findings in their manuscript fully available?

Reviewer #1: Yes

Reviewer #2: Yes

Reviewer #3: Yes

4. Is the manuscript presented in an intelligible
fashion and written in standard English?

Reviewer #1: Yes

Reviewer #2: Yes

Reviewer #3: Yes

5. Review Comments to the Author

Reviewer #1: The paper presents an evaluation of a "head joystick" (IMU-based head
motion tracker) used to control a robotic arm by a child with a congenital absence
of all four limbs as well as a control participant. They demonstrate that the
participant was able to use the head joystick to effectively control the arm and
accomplish two different tasks. Overall, I believe that the paper is well-written
and represents an appropriate first evaluation of the system. I have no major
concerns with it, and my suggestions mostly have to do with improving clarity and
informativeness.

1. Please clarify whether the participant's prior two studies with your group also
involved the head joystick.

2. Please describe the "some unstructured sessions" in more detail. How many
sessions? Approximate time per session? Approximate activities performed?

3. Please consistently put spaces between the number and unit (e.g., 0.05 m/s, not
0.05m/s).

4. While photos of the two tasks are provided, I would have appreciated some more
information about the parameters of the task - exact distances to be covered,
precision required etc.

5. I feel like more results could have been provided. In the current form, the
results are limited to task completion times and percentage of successfully
completed tasks. Would have been useful to see more information about any false
positives/negatives during switching, task completion strategies, motion smoothness,
etc. Perhaps even subjective information like user satisfaction.

Reviewer #2: The authors present a study conducted with a child with congenital limbs
absence controlling a

7dof robotic arm (Jaco, Kinova) using a head-based IMU system. The goal was to
demonstrate

that the IMU-based head joystick is well suited for the control of the robotic arm
and that the

system allowed the child to reach a level of handiness comparable to the one of an
unimpaired

individual controlling the same robotic arm with a manual joystick.

The manuscript is well organized and well written so that also non-specialists can
understand

the work. The state of the art is clear and sound. Details of the methodology are
sufficient to

allow the experiments to be reproduced and the original data are accessible.

My main concern is about the comparison between the main subject (child with
congenital limb

absence) and the control subject. If the goal is to examine the use of an IMU-based
controlling

system for a robotic arm, I think it would be more fare to compare the performance of
the

same subject using two different control systems. So in this case I would have
preferred to see

the main subject practicing with the IMU-based and with the commercially available
headcontrolled

joystick. Or have the control subject practicing and performing the tasks with
the

joystick of the Jaco and with the IMU-based joystick. My suggestion is to add a group
of healthy

subjects, not only one, that are controlling the robot with both modalities. And then
also

present the case study with the congenital lack of limbs.

Minor concerns:

- It would be nice to have an idea (on average) of how long the free exploration
during each

session was.

- There is no reference in the manuscript to Table 1.

- When the authors report the average movement time, I suggest reporting also the
standard

deviation

Reviewer #3: The paper described a case study where a child with congenital absence
of all four limbs controlled a robotic arm using custom head movement control.
However, the paper had a number of issues.

• The authors did not provide a comprehensive review of the existing work on control
interfaces for assistive robotic manipulators. Only brain control was mentioned.
JACO arms could be controlled with wheelchair joystick, head control, sip-and-puff,
and head array system. In addition, there are many types of custom control
interfaces (e.g., voice control and eye gaze control) in existing literature. It was
unclear how the proposed approach is more advantageous than existing work.

• The novelty or focus of the study is not clearly stated. From the technical
perspective, as mentioned earlier, JACO arm could be controlled with wheelchair
joystick, head control, sip-and-puff, and head array system, so it was not clear how
the head control system used in this study differed from the JACO’s existing
capability. From the clinical perspective, only one subject participated in the
study and the training/evaluation protocol was somewhat unstructured, and thus
cannot be generalized. In addition, only task completion time was reported, and user
perceived usability was not mentioned. Given the nature of such intervention, it
would be helpful to know the user feedback on the control interface and training
procedure.

• The performance contrast between the case subject and control subject is not well
justified, as the performance could be affected by not only the input devices (head
vs hand), but also their personal characteristics including but not limited to
physical limitations. Thus, it is unclear how such information would be clinically
or practically meaningful.

• The limitation of the control interface was not stated. How does head movement
control affect visual feedback a user would need for accurately controlling the arm
motion? How generalizable the approach is when used in real-life situations (e.g.,
wheelchair-mounted arm)?

6. PLOS authors have the option to publish the peer
review history of their article (what does this mean?). If published, this will
include your full peer review and any attached files.

If you choose “no”, your identity will remain anonymous but your review may still be
made public.

**Do you want your identity to be public for this peer review?** For
information about this choice, including consent withdrawal, please see our
Privacy Policy.

Reviewer #1: No

Reviewer #2: No

Reviewer #3: No

---

## [Author Response · Author response to Decision Letter 0]

14 Mar 2020

Response to Editors and Reviewers

We thank the Editor and all 3 reviewers for their insightful comments. We have
addressed these concerns with significant changes in the manuscript as seen below.
We think these changes have greatly improved the manuscript and hope that the
revised version is suitable for publication.

We have provided a point-by-point rebuttal to each comment below. For the sake of
clarity, we have color coded the text as follows:

Editors and Reviewer comments in BLACK

Authors’ response in BLUE

Corresponding changes in manuscript in GREY

-------

Journal Requirements:

We have made edits throughout our manuscript to ensure that our manuscript meets PLOS
ONE’s style requirements

We have included captions for supporting information files at the end of the
manuscript,

Supporting information

Video S1. Example drinking task for main participant. In this trial, the main
participant conducts a complete drinking task. The robotic arm is initialized to the
starting position after which the participant commands it to move towards the cup,
grasp the cup, and then move and orient the cup such that he is able to drink from
the straw.

Video S2. Example stacking task for main participant. In this trial, the main
participant conducts a complete stacking task. The robotic arm is initialized to the
starting position after which the participant commands it to approach, grasp,
orient, and place each block on the preceding block or on the table in the case of
the first block.

Data File S3. Data from participant trials as used for analysis. Included are the
movement times and dimensionless jerk values for each of the main and control
participants’ drinking and stacking trials.

3. We note that Figure 1 and your videos includes an image of a [patient /
participant / in the study]. 

We have acquired from the participant’s guardian a signed Consent Form for
Publication in a PLOS Journal and it has been mentioned in the manuscript.

The individual in this manuscript has given written informed consent (as outlined in
PLOS consent form) to publish these case details.

4. We note you have included a table to which you do not refer in the text of your
manuscript. Please ensure that you refer to Table 1 in your text; if accepted,
production will need this reference to link the reader to the Table.

We have referred to all tables present in our manuscript within the text.

Reviewer's Responses to Questions

5. Review Comments to the Author

Reviewer #1: The paper presents an evaluation of a "head joystick" (IMU-based head
motion tracker) used to control a robotic arm by a child with a congenital absence
of all four limbs as well as a control participant. They demonstrate that the
participant was able to use the head joystick to effectively control the arm and
accomplish two different tasks. Overall, I believe that the paper is well-written
and represents an appropriate first evaluation of the system. I have no major
concerns with it, and my suggestions mostly have to do with improving clarity and
informativeness.

We thank the reviewer for the positive comments

1. Please clarify whether the participant's prior two studies with your group also
involved the head joystick.

The prior studies involved control with the shoulder and torso movements, not the
head joystick. This has now been clarified in the manuscript.

These prior studies involved the control of these devices using shoulder and torso
movements.

2. Please describe the "some unstructured sessions" in more detail. How many
sessions? Approximate time per session? Approximate activities performed?

The following text has been included to describe the unstructured sessions:

Initially, there were 4 unstructured sessions, each lasting about 30-45 minutes in
length. We used these sessions to calibrate the interface to make sure that the head
movements performed were in a comfortable range when controlling the robot. During
each session, the participant was asked to do some exploratory movements of the head
to understand how movements of each DOF controlled the robot, and to learn the
operation of the switches (which was used to toggle between the
translation/orientation modes). In addition, the participant was also free to
perform any tasks of their liking using the robot arm like trying to pick up an
object from a table.

3. Please consistently put spaces between the number and unit (e.g., 0.05 m/s, not
0.05m/s).

Thank you. This has been corrected.

4. While photos of the two tasks are provided, I would have appreciated some more
information about the parameters of the task - exact distances to be covered,
precision required etc.

We thank the reviewer for this suggestion. We have now included these details in the
methods section. Table 1 has the dimensions of the stacking blocks. 

The robotic arm was always initialized in the same starting position (X: 0.40 m, Y:
0.30 m, Z: 0.30 m from the base of the robot with the gripper open enough to grasp
the cup and facing towards the right) while the cup was in front of and to the right
of the participant on the table (mean X: 0.58 m, mean Y: 0.25 m, Z: 0.00 m) - see
Fig 1c. [...] The final position of the grasper holding the cup when the main
participant took a drink was (mean X: 0.35 m, mean Y: -0.04 m, Z: 0.10 m) with the
grasper facing towards the participant.

The location of the tower was chosen by the participant (mean X: 0.39 m, mean Y: 0.28
m) to allow them to have a clear vision of the remaining blocks for the remainder of
the tasks.

Block Outer dimension of closed face (mm) Inner dimension of open face (mm)

1 44 54

2 46 61

3 53 68

4 60 77

5 67 84

Table 1. Dimensions of stacking blocks. The difference between the inner dimension of
the upper block and the outer dimension of the lower block defined the precision to
which the block had to be placed to be secure.

The blocks were placed directly across the table from the participant in five
orientations not matching the target orientation - see Fig 1d. The first block had
its opening facing upwards; the second block had its opening facing away from the
participant; the third block had its opening facing towards from the participant;
the fourth block had its opening facing to the right of the participant; the fifth
block had its opening facing to the left of the participant. These starting
positions were standardized throughout trials (position of first block with respect
to the base of the robot: mean X: 0.31 m, mean Y: 0.49 m).

5. I feel like more results could have been provided. In the current form, the
results are limited to task completion times and percentage of successfully
completed tasks. Would have been useful to see more information about any false
positives/negatives during switching, task completion strategies, motion smoothness,
etc. Perhaps even subjective information like user satisfaction.

We thank the reviewer for this suggestion. We have now included figures for the jerk
calculations (quantifying smoothness) for both drinking and stacking tasks. The
results are similar to that seen in the movement times (smoothness increases overall
with learning as evidenced by decreased jerk). 

Because the control was continuous, the only ‘discrete’ errors we could see were
during grasping (which were controlled by switches). The number of grasping errors
was quite low and we did not observe any trends with practice even as movement times
decreased. We also did not observe any major qualitative changes in task completion
strategies.

We do not have any standard measures of perceived usability or satisfaction. However,
we the fact that the participant continued this task for over 3 months and was
enthusiastic about returning for future visits is a potential indicator that he was
satisfied with the interface. We have added this in the Discussion

Moreover, although we had no direct measures of user satisfaction, the fact that the
participant continued this task for over 3 months and was enthusiastic about
returning for future visits is a potential indicator that he was satisfied with the
interface. 

These changes related to the dimensionless jerk have been incorporated in the Data
analysis and Results of the manuscript as follows:

In the Data analysis:

Dimensionless jerk. To quantify the smoothness of the movement, we computed the
dimensionless jerk values for the tasks from the end-effector position data on each
trial. The jerk was normalized by the time and peak velocity to yield a
dimensionless measure which has been shown to be more appropriate measure The
position values were first low pass filtered using a 2nd order Butterworth filter
with a cutoff frequency of 6 Hz. The jerk values for each trial were then calculated
from subsequent derivatives of the filtered position data and integrated over the
duration of the trial. The dimensionless jerk value was then computed as follows

Dimensionless Jerk= √((∫_(t_1)^(t_2)▒〖‖x‖^2 dt〗) 〖MT〗^3/v_peak^2 )

where x indicates the instantaneous jerk, MT=(t_2- t_1 ) is the movement time of
the trial, and v_peak indicates the magnitude of the peak velocity during the
trial.

In the Results:

Dimensionless jerk

Drinking task

The dimensionless jerk of the main participant closely followed the pattern of the
movement times – see Fig 7. Two trials on visit #5 (trials marked with an “x” on Fig
5) involved a lot of talking during the trial and a poorly aligned IMU.
Additionally, the data for trial #15 was corrupted and therefore omitted from the
jerk analysis. Comparing the dimensionless jerk values from the first four trials to
the last four trials shows a 45% decrease in dimensionless jerk value.

Stacking task

The dimensionless jerk of the main participant for the stacking trials also closely
followed the pattern of the movement times – see Fig 8. Comparing the jerk values
from the first three completed trials (trials 2, 5, and 8 were incomplete) to the
last three completed trials shows an 57% decrease in dimensionless jerk value.

Figure 7. Dimensionless jerk values for drinking task trials for the main participant
across practice. Jerk times denoted with a “x” were those in which the main
participant had significant distractions or poor IMU alignment which led to large
jerk values. Additionally, trial #15 is omitted due to data corruption.

Figure 8. Dimensionless jerk values for stacking task trials for the main participant
across practice. Trials 3, 5, and 8 were incomplete in which the main participant
was unable to successfully place the last block due to dropping it outside the reach
of robotic arm.

 

Reviewer #2: The authors present a study conducted with a child with congenital limbs
absence controlling a 7dof robotic arm (Jaco, Kinova) using a head-based IMU system.
The goal was to demonstrate that the IMU-based head joystick is well suited for the
control of the robotic arm and that the system allowed the child to reach a level of
handiness comparable to the one of an unimpaired individual controlling the same
robotic arm with a manual joystick.

The manuscript is well organized and well written so that also non-specialists can
understand the work. The state of the art is clear and sound. Details of the
methodology are sufficient to allow the experiments to be reproduced and the
original data are accessible.

We thank the reviewer for the positive comments

My main concern is about the comparison between the main subject (child with
congenital limb absence) and the control subject. If the goal is to examine the use
of an IMU-based controlling system for a robotic arm, I think it would be more fare
to compare the performance of the same subject using two different control systems.
So in this case I would have preferred to see the main subject practicing with the
IMU-based and with the commercially available head-controlled joystick. Or have the
control subject practicing and performing the tasks with the joystick of the Jaco
and with the IMU-based joystick. My suggestion is to add a group of healthy
subjects, not only one, that are controlling the robot with both modalities. And
then also present the case study with the congenital lack of limbs.

We thank the reviewer for the comment. We want to emphasize that the purpose of
including the control participant was not to compare the IMU interface directly with
a head joystick (there were no statistical comparisons being made), but to only
provide a baseline reference for the movement times (otherwise the magnitude of the
movement times would not be directly interpretable). 

An important novel contribution of the current study was to examine the feasibility
of the interface in children with movement impairment and show that they can
accomplish complex tasks in a reasonable period of time. Therefore, we think that
adding a group of healthy adult participants doing both interfaces (which could
provide information about the relative usability of the interfaces) is not directly
related to the current focus of the manuscript. The reviewer’s point about directly
comparing with a commercially available head joystick is well taken and while this
is not currently feasible, this line of investigation certainly lies within the
scope of our future work. 

We have now added this line to clarify the use of the control participant.

It is important to note that the control participant was an adult controlling a
manual joystick; therefore, these data are not intended to be a direct comparison
with the child using the head-joystick. Rather, given that the tasks we used were
complex tasks for which benchmarks are not already available, the data from the
control participant provide reference values that help in interpretation of the
magnitudes of the change in performance with learning and the final performance
level achieved by the child.

Minor concerns:

- It would be nice to have an idea (on average) of how long the free exploration
during each

session was.

The following text has been included:

The free exploration lasted anywhere between 3-5 minutes. The goal of the free
exploration was just to ensure that the interface was working as intended and the
participant was ready to start controlling the robot arm.

- There is no reference in the manuscript to Table 1.

We thank the reviewer for pointing this out. This is now Table 2 and is referenced in
the manuscript.

The amount and distribution of practice for both participants is shown in Table
2.

- When the authors report the average movement time, I suggest reporting also the
standard

Deviation

We have included standard deviation next to their corresponding means in the text. We
have also included the individual data points in the figures

Reviewer #3: The paper described a case study where a child with congenital absence
of all four limbs controlled a robotic arm using custom head movement control.
However, the paper had a number of issues.

• The authors did not provide a comprehensive review of the existing work on control
interfaces for assistive robotic manipulators. Only brain control was mentioned.
JACO arms could be controlled with wheelchair joystick, head control, sip-and-puff,
and head array system. In addition, there are many types of custom control
interfaces (e.g., voice control and eye gaze control) in existing literature. It was
unclear how the proposed approach is more advantageous than existing work. 

• The novelty or focus of the study is not clearly stated. From the technical
perspective, as mentioned earlier, JACO arm could be controlled with wheelchair
joystick, head control, sip-and-puff, and head array system, so it was not clear how
the head control system used in this study differed from the JACO’s existing
capability.

We thank the reviewer for raising this point. In the introduction, we compare our
method in the context of existing ‘head control’ methods (head arrays and head
joysticks) as the focus is on developing interfaces with individuals with severe
impairments (where the assumption is that that they cannot use a regular manual
joystick). Here, one main advantage of our method is to create a wireless
‘continuous’ control interface like the joystick for high DOF tasks (which is more
intuitive compared to head arrays which rely on switches).

In the discussion, we have now talked about other interfaces (such as the sip and
puff system, eye gaze control, voice commands). One primary advantage of our
interface is that it is designed to be flexible and used for high-DOF control with
precision. The relative advantages of each system will depend to a great extent on
the abilities of the individual (e.g. a more severely impaired individual with
limited head movements may benefit from a voice controlled or sip-and-puff system).
We have included this text in the Discussion

In addition to head control methods discussed here, several alternate control
interfaces to manual joysticks have been developed for individuals with severe
movement impairments. These include sip-and-puff systems, voice control [23], gaze
control [24] and tongue control [25]. These interfaces typically involve some
tradeoff between (i) the number of control dimensions (e.g., a device that only
allows control of 1 or 2 dimensions would require frequent ‘switching’ to control a
high DOF robotic arm), (ii) the type of control (e.g., discrete controls are
possible using voice commands but are less intuitive and precise relative to
continuous control like a joystick), and (iii) the ‘invasiveness’ of the device both
in terms of its physical attributes (e.g., whether it is easily wearable, wireless
etc.) but also how it affects other activities such as communication (e.g., gaze or
voice based controls may interfere with natural day-to-day behavior). Ultimately,
the choice of the interface will depend both on existing movement abilities for the
individual and the number of degrees of freedom to be controlled.

From the clinical perspective, only one subject participated in the study and the
training/evaluation protocol was somewhat unstructured, and thus cannot be
generalized.

This is a highly unique population (child with no movement of the limbs)- so the goal
was to provide proof of concept that the head IMU interfaces are feasible for
complex movements in children. We have added this as a limitation of the study.

Although conducted as a case study, which places limits on generalizability, our
results add to these prior findings by demonstrating that (i) these interfaces are
well-suited for children, and (ii) the improvement in performance over multiple
practice sessions is substantial (up to 40-50% reduction in movement times) and
comparable to manual joystick performance.

In addition, only task completion time was reported, and user perceived usability was
not mentioned. Given the nature of such intervention, it would be helpful to know
the user feedback on the control interface and training procedure.

The reviewer makes a good point but we do not have any standard measures of perceived
usability or satisfaction. However, the fact that the participant continued this
task enthusiastically for over 3 months is an indicator that he was satisfied with
the interface. 

• The performance contrast between the case subject and control subject is not well
justified, as the performance could be affected by not only the input devices (head
vs hand), but also their personal characteristics including but not limited to
physical limitations. Thus, it is unclear how such information would be clinically
or practically meaningful.

As in our response to reviewer 2, the performance of the control subject was only to
provide a baseline reference for the movement times (and not to directly compare
them). We have now added this line to clarify the use of the control
participant.

Given that the control participant was an adult controlling a manual joystick, these
values are not intended to be a direct comparison with the child using the
head-joystick. Rather, given that the tasks we used were complex tasks for which
benchmarks are not already available, the data from the control participant provide
reference values that help interpretation of the magnitudes of the change in
performance over learning and the final performance level achieved by the child.

• The limitation of the control interface was not stated. How does head movement
control affect visual feedback a user would need for accurately controlling the arm
motion? How generalizable the approach is when used in real-life situations (e.g.,
wheelchair-mounted arm)?

Both tasks required visual feedback since they required precision to hold the cup or
stack the blocks. The head movements performed were quite small; because head
movements directly controlled the velocity of the robot, big movements of the head
were not needed. Additionally, when big movements of the robot were needed (where
visual feedback is not as critical). So at least to a first approximation, these
movements did not seem to affect the use of visual feedback to control the movement
of the robot arm. Even when it is mounted on a wheelchair, it is unlikely that the
user will do both wheelchair navigation and robot arm control simultaneously;
therefore, we do not think that the head movements will pose a problem in real-world
situations. 

We have referred to this in the Discussion

It is also worth noting that despite requiring the use of head movements to control
the robot to precisely control the end effector, the child was able to successfully
perform these tasks, indicating that the small head movements performed did not
interfere greatly with the use of visual feedback..

We refer to the limitations of the current approach in the penultimate paragraph in
the Discussion. (Burden of learning being completely on the user, and the exclusive
use of head signals)

In terms of further improvements to our design, we wish to highlight two issues.
First, a limitation of our approach is that the ‘burden of learning’ is all on the
user. This may be especially challenging for children, who show deficits relative to
adults in learning such interfaces [21], [26]. One way to improve this is to use
either an adaptive interface that adjusts to the user [27], [28], or use a shared
control framework so that the autonomy of control can be shared between the human
and the machine [29]. Second, for the sake of simplicity, we relied only on head
movements (i.e. kinematics) to control the device. However, in the control of other
neuroprosthetics, electromyographic signals from different muscles is often used to
augment the movement repertoire by providing distinct control signals for the
control of the external device [30]–[32]. Therefore, a hybrid combination of IMU
signals along with electromyography may further facilitate the user for efficient
control of high DOFs [33].

We have also added this line to highlight the goal of responding in real-life
situations

Addressing these limitations could increase the potential of this approach to deal
with real-life situations which require both speed and accuracy.

Review Response.docx
---

## [Decision Letter · Decision Letter 1]

29 Apr 2020

PONE-D-19-31845R1

Controlling a robotic arm for functional tasks using a wireless head-joystick: A case
study of a child with congenital absence of upper and lower limbs

PLOS ONE

Dear Mr. Aspelund,

Thank you for submitting your manuscript to PLOS ONE. After careful consideration, we
feel that it has merit but does not fully meet PLOS ONE’s publication criteria as it
currently stands. Therefore, we invite you to submit a revised version of the
manuscript that addresses the points raised during the review process.

We would appreciate receiving your revised manuscript by Jun 13 2020 11:59PM. When
you are ready to submit your revision, log on to https://www.editorialmanager.com/pone/ and select the 'Submissions
Needing Revision' folder to locate your manuscript file.

If you would like to make changes to your financial disclosure, please include your
updated statement in your cover letter.

To enhance the reproducibility of your results, we recommend that if applicable you
deposit your laboratory protocols in protocols.io, where a protocol can be assigned
its own identifier (DOI) such that it can be cited independently in the future. For
instructions see: http://journals.plos.org/plosone/s/submission-guidelines#loc-laboratory-protocols

We look forward to receiving your revised manuscript.

Kind regards,

Imre Cikajlo, Ph.D.

Academic Editor

PLOS ONE

Additional Editor Comments (if provided):

Please carefully examine the manuscript with reviewers' comments in mind. Thank
you.

Reviewers' comments:

Reviewer's Responses to Questions

**Comments to the Author**

1. If the authors have adequately addressed your comments raised in a previous round
of review and you feel that this manuscript is now acceptable for publication, you
may indicate that here to bypass the “Comments to the Author” section, enter your
conflict of interest statement in the “Confidential to Editor” section, and submit
your "Accept" recommendation.

Reviewer #1: All comments have been addressed

Reviewer #2: (No Response)

2. Is the manuscript technically sound, and do the data
support the conclusions?

Reviewer #1: (No Response)

Reviewer #2: Yes

3. Has the statistical analysis been performed
appropriately and rigorously? 

Reviewer #1: (No Response)

Reviewer #2: N/A

4. Have the authors made all data underlying the
findings in their manuscript fully available?

Reviewer #1: (No Response)

Reviewer #2: Yes

5. Is the manuscript presented in an intelligible
fashion and written in standard English?

Reviewer #1: (No Response)

Reviewer #2: Yes

6. Review Comments to the Author

Reviewer #1: (No Response)

Reviewer #2: I want to tank the authors for the detailed replies to all my concerns.
Everything is good with the exception of the main concern. I am still not fully
convinced about the lack of a control group. I understand that comparing the
IMU-based interface with the commercial available interface is out of the scope of
this specific paper, and I am okay with that. As the author said themselves “the
purpose of including the control participant was to only provide a baseline
reference for the movement times” but with n=1 the baseline values might not be true
but instead a “false positive”. My suggestion is to include data of few healthy
subjects.

7. PLOS authors have the option to publish the peer
review history of their article (what does this mean?). If published, this will
include your full peer review and any attached files.

If you choose “no”, your identity will remain anonymous but your review may still be
made public.

**Do you want your identity to be public for this peer review?** For
information about this choice, including consent withdrawal, please see our
Privacy Policy.

Reviewer #1: No

Reviewer #2: No

---

## [Author Response · Author response to Decision Letter 1]

2 Jun 2020

Response to Editors and Reviewers

We thank the Editor and the reviewers for their insightful comments. We have
addressed these concerns with changes in the manuscript as seen below. We think
these changes have improved the manuscript and hope that the revised version is
suitable for publication.

We have provided a point-by-point rebuttal to each comment below. For the sake of
clarity, we have color coded the text as follows:

Editors and Reviewer comments in BLACK

Authors’ response in BLUE

Corresponding changes in manuscript in GREY

-------

Journal Requirements:

N/A

Reviewer's Responses to Questions

6. Review Comments to the Author

Reviewer #1: (No Response)

Reviewer #2: I want to tank the authors for the detailed replies to all my concerns.
Everything is good with the exception of the main concern. I am still not fully
convinced about the lack of a control group. I understand that comparing the
IMU-based interface with the commercial available interface is out of the scope of
this specific paper, and I am okay with that. As the author said themselves “the
purpose of including the control participant was to only provide a baseline
reference for the movement times” but with n=1 the baseline values might not be true
but instead a “false positive”. My suggestion is to include data of few healthy
subjects.

We thank the reviewer for the comment. We understand the reviewer’s concerns
regarding the fact that we had only a single individual as control (although we had
multiple measurements over multiple visits on this individual and we provide an
indication of not only the mean but the full range of values seen during practice).
Also, anecdotally we wish to note that the best times for the stacking task of the
control participant and the main participant were within 25% of the best time of
what can be considered an expert user’s time (the expert user was the researcher who
developed the system and had practiced on it extensively during testing) – main
best: 446 s, control best: 376 s, expert best ~300 s. Even though this data is
anecdotal, we think it gives us confidence that the data we report are not far off
the mark.

Ultimately, given this is a case-study, we think this result can only be treated as a
proof of concept and is not intended to be generalizable - so issues such as false
positives or negatives are not applicable. Moreover, given the current situation
with COVID-19, we believe that we will not be able to get this data for several
months, and it would not fundamentally alter any of the conclusions in the paper. 

We have now explicitly addressed the limitations of our results as needing further
study in order to be applied generally. We hope that this additional narrowing of
the scope of the generality of our work is sufficient.

"Our results add to these prior findings by demonstrating that (i) these interfaces
are well-suited for children, and (ii) the improvement in performance over multiple
practice sessions is substantial (up to 40-50% reduction in movement times). The
child’s performance for the tasks was found to be comparable to that of an adult
control participant using a manual joystick. However, given that we only had data
from a single child and a single adult, additional studies are needed for assessing
the generality of these findings."

Response to Reviewers 2.docx
---

## [Editor Report · Decision Letter 2]

30 Jun 2020

Controlling a robotic arm for functional tasks using a wireless head-joystick: A case
study of a child with congenital absence of upper and lower limbs

PONE-D-19-31845R2

Dear Dr. Aspelund,

We’re pleased to inform you that your manuscript has been judged scientifically
suitable for publication and will be formally accepted for publication once it meets
all outstanding technical requirements.

Kind regards,

Imre Cikajlo, Ph.D.

Academic Editor

PLOS ONE
---

## [Editor Report · Acceptance letter]

13 Jul 2020

PONE-D-19-31845R2 

Controlling a robotic arm for functional tasks using a wireless head-joystick: A case
study of a child with congenital absence of upper and lower limbs 

Dear Dr. Aspelund:

I'm pleased to inform you that your manuscript has been deemed suitable for
publication in PLOS ONE. Congratulations! Your manuscript is now with our production
department. 

Kind regards, 

on behalf of

Professor Imre Cikajlo 

Academic Editor

PLOS ONE